# *CfHMG* Differentially Regulates the Sexual Development and Pathogenicity of *Colletotrichum fructicola* Plus and Minus Strains

**DOI:** 10.3390/jof10070478

**Published:** 2024-07-11

**Authors:** Wei Zhang, Wenkui Liu, Xiaofei Liang, Rong Zhang, Mark L. Gleason, Guangyu Sun

**Affiliations:** 1State Key Laboratory of Crop Stress Biology in Arid Areas and College of Plant Protection, Northwest A&F University, Yangling 712100, China; weizhangwzh@163.com (W.Z.); liuwenkui@nwsuaf.edu.cn (W.L.); rongzh@nwsuaf.edu.cn (R.Z.); 2Department of Plant Pathology and Microbiology, Iowa State University, Ames, IA 50011, USA; mgleason@iastate.edu

**Keywords:** *Colletotrichum*, HMG transcription factor, pathogenicity, conidiation, mating type

## Abstract

*Colletotrichum fructicola* shows morphological and genetic differences in plus and minus strains. However, the mechanism of the differentiation between two types of strains is still largely unclear. Our early transcriptome analysis revealed that *CfHMG* expression differed in plus and minus strains. To define the functions of the *CfHMG* gene, we constructed gene deletion mutants by homologous recombination. We found that a *CfHMG* deletion mutant of the minus strain, CfHMG-M, could lead to a reduction in perithecium sizes and densities on media and sterile perithecium formation compared with the minus wild type (WT), whereas there was no effect for the plus mutant CfHMG-P. In co-cultures between CfHMG-P and minus WT, CfHMG-M and plus WT, or CfHMG-P and CfHMG-M, the quantities of perithecia were all reduced significantly. When conidial suspensions were inoculated on non-wounded apple fruit, it was found that the virulence of the minus mutant decreased significantly but not for the plus one. Further, we found that the virulence decrease in minus mutants was caused by a decrease in the conidium germination rate. Our results indicate that *CfHMG* of *C. fructicola* plays an important role in the mating line formation between the plus and minus strain for both strains and differentially regulates the perithecium size, density, fertilization, and virulence of the minus strain. The results are significant for further detecting the differentiated mechanisms between the plus and minus strains in *Colletotrichum* fungi.

## 1. Introduction

*Colletotrichum* species are among the 10 most damaging genera of fungal plant pathogens in the world [1,2]. Glomerella leaf spot (GLS) caused by *Colletotrichum* spp. is a devastating disease in apple (*Malus* × *domestica*), resulting in severe premature defoliation and fruit lesions [3]. GLS was first found in southeast USA [4]. Later, it was found in Brazil, China, Japan, and Uruguay [5,6,7,8]. *Colletotrichum fructicola* is the main pathogen of GLS in China, in addition to apple, *Colletotrichum fructicola* also damages many commercial crops, such as pear, strawberry, tea, rubber trees, and some ornamental plants [9,10,11,12,13].

Edgerton [14,15] differentiated the teleomorph of *Colletotrichum* into plus and minus types according to differences in morphology and sexual reproduction. The plus strain usually develops grey and livid colonies with thick and felty aerial mycelia, accompanied by aggregated masses of fertile perithecia with asci and ascospores. In contrast, the minus strains are black colonies with sparse aerial mycelia, sparse perithecia that produce a few or no asci. Plus and minus strains are sexually compatible each other. On artificial media, a mating line forms on the intersection of plus- and minus-strain co-culture colonies, along with abundant fertile perithecia. Dong et al. [16] found that there were plus and minus strains in *C. fructicola*. Kong et al. [17] found that the autophagy gene *CfAtg8* of *C. fructicola* was differentially expressed in the plus and minus strains and that a C2H2-containing transcription factor *Cfcpmd1* was also found related to plus and minus differentiation [18]. At present, however, the differentiation mechanism of plus and minus remains elusive.

High-mobility-group box (HMGB) proteins belong to the HMG superfamily, a large group of transcription factors, which is divided into the UBF_HMG and SOX/TCF/MATA_HMG family [19]. The MATA_HMG subfamily encodes proteins relevant to fungi that are important to regulate differentiation and the sexual process. Most of the genes in this subfamily are located in the mating type locus and act as mating genes, whereas others are located outside of the mating type gene cluster [20].

In our previous whole genomics and transcriptome analysis, a MATA_HMG protein, CfHMG, was found to be expressed differently in the plus and minus strains of *C. fructicola* [21]. In the present study, we found that *CfHMG* differentially regulated the pathogenicity, sexual reproduction, and mating in plus and minus strains of this fungus and clarified the mechanism of the differentiation of plus–minus phenotypes.

## 2. Materials and Methods

### 2.1. Fungal Strains, Media, Culture Conditions

Wild-type 1104-4 (plus) and 1104-6 (minus) strains of *Colletotrichum fructicola* Prihastuti, L. Cai & K.D. Hyde from apple leaves with GLS were stored in the laboratory of the Fungal Research Laboratory of NWAFU, Yangling, China. Cultures were grown on potato dextrose agar (PDA) or oatmeal agar (OA) at 25 °C. For mating line formation or cross-fertility tests, both strains were co-cultured on OA media [22]. The mycelial plugs of strains were preserved in 15% glycerol at −80 °C for long-time storage. The mycelial plugs of strains were preserved in 15% glycerol at −80 °C for long-term storage. To obtain conidia, three mycelial plugs with a diameter of 6 mm were placed in 60 mL of potato dextrose broth (PDB) to produce conidia with shaking at 180 rpm at 25 °C for 4 d, after which the culture liquids were filtered by three layers of sterile MiraCloth (EMD Millipore Corporation, Burlington, MA, USA). The filtrate was centrifuged for 3 min at 10,000 rpm and 4 °C, and then, the supernatant was discarded.

### 2.2. Sequence Analysis, Deletion, and Complementation of CfHMG

The *CfHMG* gene sequence was obtained from *C. fructicola* 1104-7 genome (GenBank Accession No. MVNS00000000.2). CfHMG homolog animo acid sequences in fungal species from *Colletotrichum* spp. and other species were searched by the blastp program on NCBI. DNAman 6.0 and MEGA 7.0 were used for sequence alignment and phylogenetic analysis.

Genomic DNA (gDNA) of aerial hyphae was extracted by the cetyltrimethylammonium bromide (CTAB) method, and the *CfHMG* deletion mutants were generated using a split-marker approach [23]. The upstream and downstream flanking sequences of *CfHMG* were amplified with primers CfHMG-LFup/CfHMG-LRup and CfHMG-RFDown/CfHMG-RRDown from gDNA. The gene replacement cassette was constructed by connecting upstream and downstream flanking sequences with hygromycin resistance gene. Primers CfHMG-LF/HyR and NYGF/CfHMG-RR were used to amplify two split-marker-based gene replacement constructs. The final PCR products were purified and transformed to the wild-type strains by PEG-mediated protoplast transformation [24]. To confirm the gene deletion strains, three pairs of primers were used to detect wild types and *CfHMG* gene deletion mutants. The fragments of upstream and downstream amplified by LF/Xu855R and Xu866F/RR were only observed in the *CfHMG* deletion mutants. In addition, a fragment amplified by DF/DR at the ORF region of the *CfHMG* gene in WT strain was missing in *CfHMG* deletion mutants (Table A1).

To complement *CfHMG*-deletion strains, recombinant pHZ-100-CfHMG was brought into protoplasts of mutant strains Δ*CfHMG-P* and Δ*CfHMG-M*. Complementation strains were identified from G418-resistant transformants using PCR.

### 2.3. Tests for the Perithecium Development and Mating Line Formation between Plus and Minus Strains

For analyzing the gene functions of *CfHMG* in sexual development, isolates of wild types and mutants were tested for both self-fertility and cross fertility in all possible co-culture combinations on OA media.

For self-fertility tests, 5 mm diameter plugs taken from a 3-day-old PDA colony were transferred to OA and cultured for a further 7 d at 25 °C in darkness. For WT of minus strains, mutant Δ*CfHMG-M*, and complement strain Δ*CfHMG-MC*, OA plugs with perithecia were examined microscopically after 7 d to estimate the time to formation as well as the density and the size of perithecia. More than 100 perithecia were measured. Meanwhile, for WT of plus strains, mutant Δ*CfHMG-P*, and complement strain Δ*CfHMG-PC*, the number of perithecia per cluster in each plate of diameter 9 cm were counted after 7 dpi, and their diameters were measured. For cross-fertility-tested strains, OA medium plugs from cross lines were sampled after 2 dpi of contact between different strains for the microscopical examination of perithecia.

### 2.4. Pathogenicity Assays

To analyze the ability to colonize apple fruit, conidial suspensions (20 μL) of deletion mutant strains and their associated wild-type strains were inoculated in the non-wounded fruit of *Malus domestica* Borkh. cv. Gala. To clearly see the lesions on the infected fruit, the bagged mature apple fruit in yellow were used. Conidial suspension was prepared with shaken cultures in PDB medium at 25 °C and then adjusted to a concentration of 5 × 10^5^ conidia/mL in sterile distilled water. Apple fruit surfaces were sterilized by rubbing with 70% ethanol, followed by wiping with sterile distilled water. The fruits were sprayed with the conidial suspensions and then incubated at 25 °C in a moisture chamber. Photos were taken, and the diameter of each lesion was measured after 6 d. Three fruits were used per strain in each run of the assay. All inoculations were conducted at least three times independently using all the wild-type strains and *CfHMG* mutant strains, and similar results were obtained.

### 2.5. Measurements of Conidial Germination, Appressorium Formation

To estimate the rate of appressorium formation and the development of infection hyphae, conidial suspensions of each strain were spread on cellophane appressed to the surface of water–agar medium and incubated at 25 °C in darkness. Amounts of appressoria formation and infection hyphae were assessed at 12 and 16 h, respectively.

## 3. Results

### 3.1. CfHMG Encoded a Novel MATA_HMG Protein

The *CfHMG* was found to be a single-copy gene in *C. fructicola* (both plus and minus strain) genome based on a local blast search. Sequence analysis revealed that *CfHMG* carried an open reading frame (ORF) of 2151 bp containing two exons and one intron. The gene encoded a putative HMG protein of 682 amino acids containing a highly conserved high-mobility-group box. The sequence homolog bast of CfHMG showed that there was very high aa identity in the *Colletotrichum* (the identity > 67.5%) and higher similarity in Gloeosporioides section (the identity > 96.5%). All *Colletotrichum* spp. were clustered in a clade in the phylogenetic tree of whole CfHMG aa sequences of *Colletotrichum* spp. and related genera (Figure 1A). A further aa sequence comparison of the HMG motif showed that it was conserved in *Colletotrichum* spp. (100% identity) and with high similarity with *Fusarium oxysporium*, *Verticillium longisporium*, *Hypoxylon fuscum*, etc. However, there were lower sequence similarities with Mat 1-2-1 and other function-known proteins, including *Saccharomyces cerevisiae* Rox, *Candida albicans* Rfg, *Podospora anserina* PaHMG5, and *Schizosacharomyces pombe* Ste11 (Figure 1B).

### 3.2. Generating Gene Deletion and Complementation Mutants of CfHMG

For the functional characterization of *CfHMG*, we generated gene deletion mutants by a PEG-mediated transformation of the gene replacement cassette to the plus and minus wild-type strains (Figure 2A). Putative deletion strains were identified by PCR amplification with different primers (Table A1). We obtained mutant Δ*CfHMG-P* from plus wild type WT(P) and mutant Δ*CfHMG-M* from minus wild type WT(M). Each deletion mutant was purified by the single spore purification procedure. In the meantime, we obtained complementation strains Δ*CfHMG-PC* and Δ*CfHMG-MC*. To test the function of *CfHMG* in hyphal growth, we observed colony morphology and growth rate on PDA. The mutants showed similar growth rates to wild types, and no obvious difference was observed in colony morphology between wild types and mutants (Figure 2B).

### 3.3. CfHMG Affected the Development and Fertility of Perithecium in the Minus Strain

To illustrate the function of *CfHMG* in sexual reproduction, the strains were cultured on OA media, and the perithecium amount, size, and fertility were observed. The results showed that WT(M) produced abundant, scattered, dark black perithecia; among them, about 10% perithecia could produce asci and ascospores (Figure 3A–D). In contrast, the perithecia of the knockout mutant Δ*CfHMG-M* had an average diameter 45.1 ± 6.4 μm, which was much smaller than that in WT(M) (70.9 1 ± 7.1 1 μm) (Figure 3A,B). In addition, we found that the deletion of *CfHMG* caused a sharp increase in the density of perithecia when compared with WT(M) (Figure 3A,C), and no ascus and ascospore was found in mutants (Figure 3A,D).

For the wild plus strain WT(P), deletion of *CfHMG* mutant, and complement mutant Δ*CfHMG-PC*, all strains could produce perithecial cluster and fertile perithecia (Figure 4), implying that *CfHMG* did not affect the sexual reproduction structure and the fertility for the plus strain, which was different from that in the minus strain.

### 3.4. CfHMG Is Involved in the Formation of Mating Line for Both Plus and Minus Strains

To further analyze the functions of *CfHMG* in sexual reproduction, tests were arranged for cross-fertility in various combinations. For the WT(P) and WT(M) co-culture, a nigrescent line appeared in the contact area after contact for about 3 d, and a clear mating line appeared within 7 d. Lots of scattered or clumped perithecia formed on the line (Figure 5A). Compared with the wild strains, the number of perithecia on the mating line of Δ*CfHMG-P* × WT(M) co-culturing decreased 60.0%; for Δ*CfHMG-M* × WT(P) co-culturing, it decreased about 34.8%. For Δ*CfHMG-P* × Δ*CfHMG-M*, fewer perithecia formed, which decreased about 96.2% compared with the wild strains (Figure 5B). In addition, for the co-culturing of Δ*CfHMG-P* × Δ*CfHMG-M*, the average diameter of perithecia was 38.7 μm, which was smaller than the wild strains (62.0 μm) (Figure 5C). These results suggest that *CfHMG* could affect the mating line formation ability of both plus and minus strains.

### 3.5. CfHMG Deletion Reduces Virulence in Minus Strain

To investigate if *CfHMG* influences the virulence of *C. fructicola*, we assayed the pathogenicity of WT(P), WT(M), and relevant mutants on apple fruit. The results showed that the disease index of *CfHMG-M* reduced by approximately 25% compared with the wild mutant WT(M), and *CfHMG-MC* could restore most of the virulence which the deletion mutant lost, while there was no difference between WT(P) and their mutants (Figure 6). These results suggest that *CfHMG* played an important role in virulence in *C. fructicola* in the minus strain but not in the plus strain.

To further understand the reason for the *CfHMG* mutant losing virulence, the infection-related structures were observed on cellophane. It was found that the conidial germination rate of wild strain WT(M) was 82.7%, and the germination rate of Δ*CfHMG-M* was only 52.4%. Compared to WT(M), it decreased about 36.6% (Figure 7B), while for the plus strain, there was no significant difference between the wild and mutants, implying that *CfHMG* did not affect the conidium germination for the plus strain. In addition, there was almost no difference among the wild type and mutants in both plus strains or minus strains in rates of appressorium formation and penetration hypha. These results imply that the virulence reduction might be caused by a decrease in conidial germination rate in the minus strain.

## 4. Conclusions

In this study, we characterized the functions of an HMG-BOX transcription factor *CfHMG* in *Colletotrichum fructicola* by loss-of-function and gain-of-function analysis. We found that *CfHMG* could affect the sizes, densities on media, and fertility of perithecia for the minus strain, whereas there was no effect for the plus one. The co-culture of plus and minus strains assay showed that *CfHMG* played an important role in the mating line formation on media. It was also found that *CfHMG* deletion could reduce the virulence of the minus mutant but not of the plus one. Furthermore, the percentage of conidium germination was lower, implying that *CfHMG* regulates the virulence of the minus strain possibly by affecting its conidium germination.

## 5. Discussion

The differentiation of plus and minus strains was first described in *Glomerella cingulata*, the teleomorph of *C. gloeosporioides* [25]. In the recent classification system, former *C. gloeosporioides* was thought of as a species complex, and Gloeosporioides section. *C. fructicola* was identified as a unique species in this section [9,26]. Liang et al. [22] found that the expression levels of pheromone precursors and receptors in *C. fructicola* were different in the mycelia of two type strains and supposed this might be a reason for the differentiation of plus and minus strains. Kong et al. [17] found that the absence of autophagy gene *CfAtg8* had different effects on the colony morphology and mating of plus and minus strains, indicating that this gene is an important regulator of the differentiation of plus and minus strains. In addition, the C2H2 transcription factor *CfCpmd1* was also found to be involved in the regulation of plus and minus strain differentiation [18]. In our study, we found that *CfHMG* played an important role in differentially regulating perithecium development, crossing ability, and pathogenicity in *C. fructicola* plus and minus strains, implying that *CfHMG* is one of the key genes regulating differentiation in *C. fructicola*.

The MATA_HMG subfamily transcription factors that determine the sexual differentiation of fungi and regulate the sexual reproduction process occur widely among fungi. Some members of the MATA_HMG subfamily regulate differentiation and the sexual process directly, such as the most studied mating type gene *MAT1-2-1*. Phylogenetic analysis showed that CfHMG protein formed a monophyletic clade within the genus *Colletotrichum* and that these amino acid sequences were highly conserved, indicating that CfHMG orthologs in *Colletotrichum* may have similar functions. Phylogeny analysis and amino acid comparisons showed that CfHMG homology proteins share lower similarity with *MAT1-2-1*, even in the HMG-box motif region. Our genome annotation also found that CfHMG did not distribute in the mating type gene cluster, indicating CfHMG is different from MAT1-2-1.

In *Schizosaccharomyces pombe*, the *fmf* gene encodes a MATA_HMG-box protein SpSte11, which regulates the transcription of mating genes *matP* and *matM* through nuclear accumulation, thus regulating the sexual reproduction process [27,28]. In *P. anserina*, the PaHMG5 is a key activator of mating type genes and is located at the center of several HMG-box factor networks that regulate mating type genes and mating type target genes [29]. ROX1 in *S. cerevisiae* inhibits the expression of hypoxia genes [30]. In *Candida albicans*, Rfg1 controls filamentous growth in an environment-dependent manner and affects virulence in mice [31]. In our study, the knockout of *CfHMG* resulted in defective perithecia development, impaired virulence in the minus strain, and the possible recognition between the minus and plus strain. The amino acid comparisons of the HMG-box motif region showed that CfHMG shares lower homology with Rfg1, ROX1, Ste11, and PaHMG5. Our finding showed for the first time that CfHMG was a new functional protein different from the reported MATA_HMG-box proteins.

Conidial germination is critical for *Colletotrichum* fungi to infect hosts. Many factors affect the germination of conidia. Some genes affecting conidial germination of *C. fructicola* have also been found, such as ubiquitin-binding enzyme protein Chip1, vacuole-copper ion transporter CgCTR2 [32,33]. In our study, the deletion of *CfHMG* caused reduction in conidial germination and further decreased the virulence of the minus strain.

Differentiating plus and minus strains is a very important property of the *Colletotrichum* fungi. However, because both plus and minus strains can carry out the relatively less efficient sexual reproduction, the more efficient sexual reproduction occurs when plus and minus strains meet. Our research found *CfHMG* was related to the differentiation of plus and minus strains and differentially regulated the perithecium development, mating ability, and virulence of *C. fructicola*, providing valuable information for understanding the regulatory mechanisms of plus–minus differentiation. Considering the economic loss caused by *Colletotrichum* fungi and the high sequence conservation of CfHMG in *Colletotrichum*, *CfHMG* could be treated as a potential fungicidal target for designing novel fungicides in the future.

## Figures and Tables

**Figure 1 jof-10-00478-f001:**
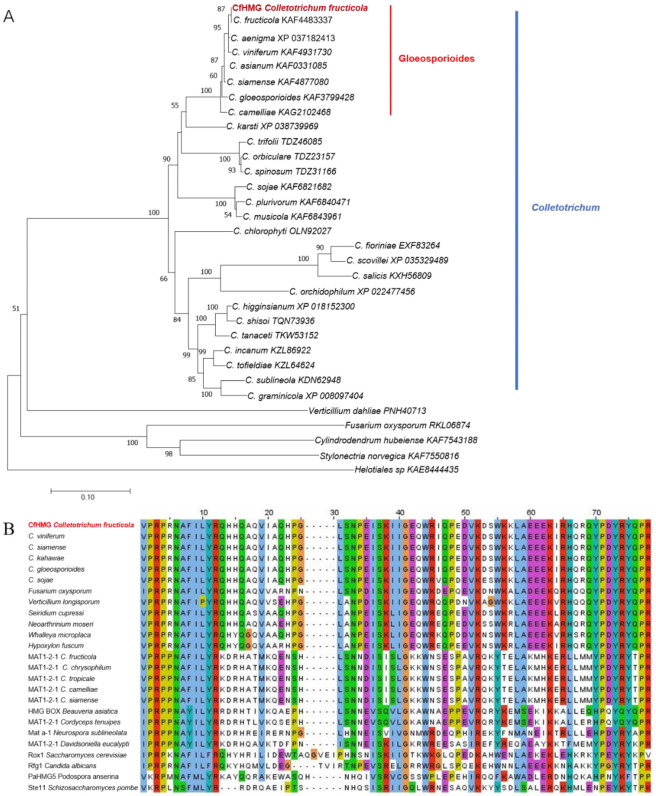
NJ phylogenetic tree of homolog proteins and sequence comparison of CfHMG among *Colletotrichum* and related genera. (**A**) Phylogenetic tree of CfHMG and its homolog proteins in *Colletotrichum*. Amino acid sequences were analyzed by MEGA 5 using a neighbor-joining bootstrap (1000 replicates). The scale bar corresponds to a genetic distance of 0.10. (**B**) Sequence comparison of the HMG box from CfHMG, MAT1-2-1, and function-known proteins. The alignment was performed using—ClustalX 2.1 and color scheme provided by Jalview 2.10.5.

**Figure 2 jof-10-00478-f002:**
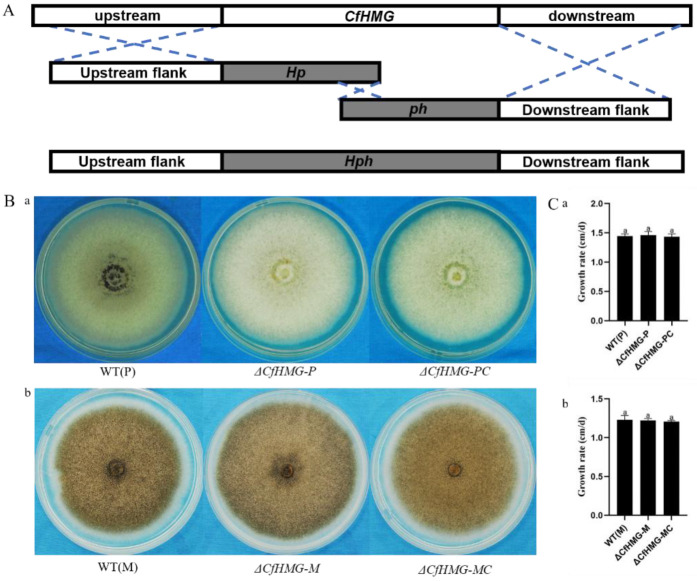
*CfHMG* deletion mutants showed no differences in vegetative growth and colony morphology for both plus and minus strains. (**A**) Strategy for generating the *CfHMG* gene deletion mutants. Black arrows represent primer, and black box (Hp, ph) are two split fragments of the hygromycin phosphotransferase gene. (**B**) Colony morphology of strains on potato dextrose agar for 7 d at 25 °C. (a): Plus wild type, mutant Δ*CfHMG-P*, and complementation Δ*CfHMG-PC*. (b): Minus wild type, mutant ΔCfHMG-M, and complementation Δ*CfHMG-MC*. (**C**) Growth rate comparison. The mean and standard deviation were calculated from three biological replicates. Different letters in (**C**) represent significant differences (*p* < 0.05) based on one-way ANOVA followed by post-hoc Tukey test.

**Figure 3 jof-10-00478-f003:**
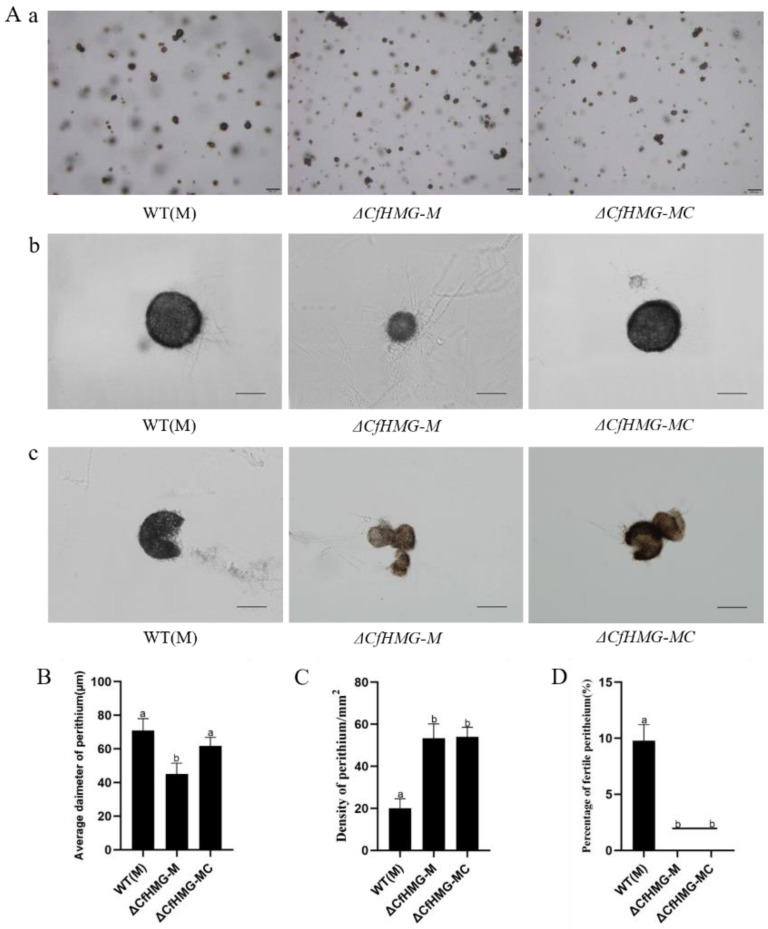
*CfHMG* deletion led to defects in the development of perithecia in minus strains on OA medium. (**A**) (a): Perithecium formation. Bar = 200 μm; (b): Perithecium size. Bar = 50 μm. (c). Crushed perithecia. Bar = 50 μm. (**B**) Average diameter comparison of perithecia. (**C**) The density comparison of perithecia. (**D**) The percentage of fertile perithecia. Different letters in (**B**–**D**) represent significant differences (*p* < 0.05) based on one-way ANOVA followed by post-hoc Tukey test.

**Figure 4 jof-10-00478-f004:**
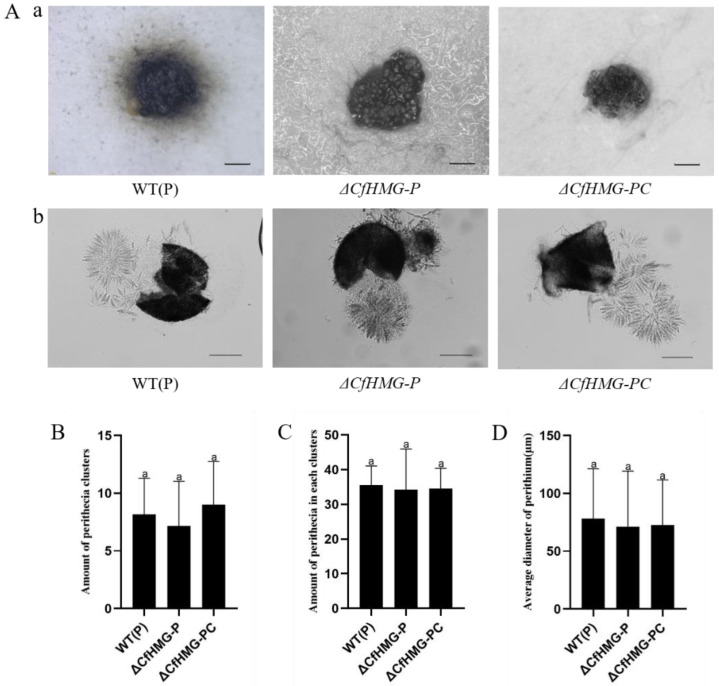
*CfHMG* deletion does not affect the development of perithecia in plus strains. (**A**) (a): Perithecial cluster. Bar = 50 μm. (b): Asci, ascospores, and perithecia. Bar = 50 μm. (**B**) Amount of perithecial cluster per plate. (**C**) Amount of perithecia per perithecial cluster. (**D**) Average diameter of perithecia. Different letters in (**B**–**D**) represent significant differences (*p* < 0.05) based on one-way ANOVA followed by post-hoc Tukey test.

**Figure 5 jof-10-00478-f005:**
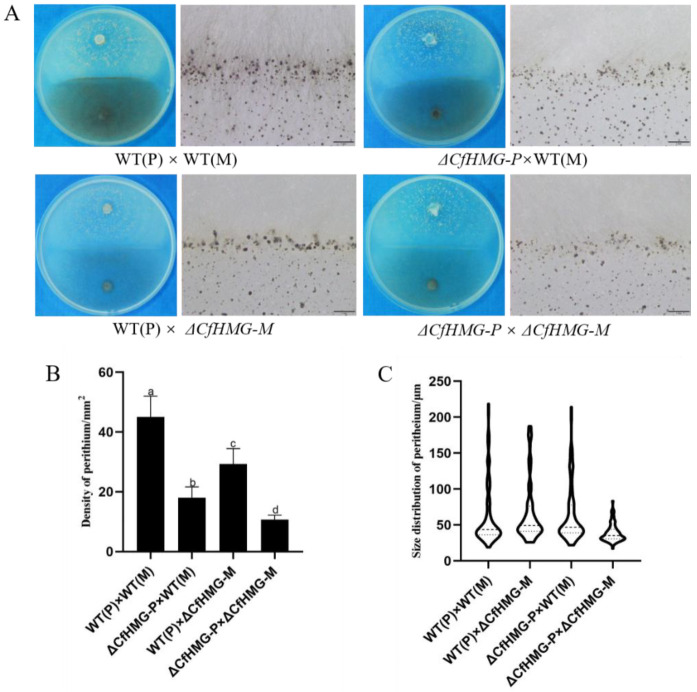
*CfHMG* deletion affects mating ability for plus and minus strains. (**A**) Mating line formation induced by co-culturing with WT(P), WT(M), and relevant mutants on OA. Bar = 1 mm. (**B**) The comparisons of perithecium density on mating line on OA. (**C**) Size distribution comparisons of perithecia on mating line formation induced by co-culturing on OA. Different letters in (**B**) represent significant differences (*p* < 0.05) based on one-way ANOVA followed by post-hoc Tukey test.

**Figure 6 jof-10-00478-f006:**
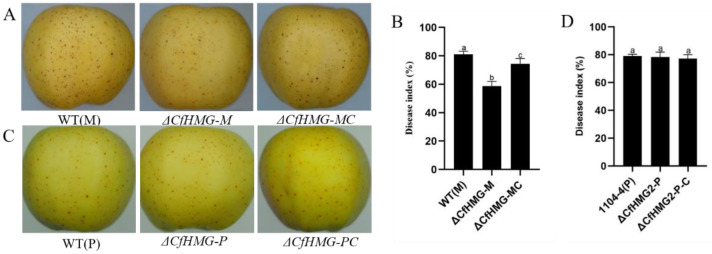
Deletion of *CfHMG* reduced the virulence in the minus strain. (**A**) Non-wounded inoculation assays of minus strains on apple. (**C**) Non-wounded inoculation assays of plus strains on apple. (**B**) Disease index comparison among minus and mutant strains.(**D**) Disease index comparison among plus and mutant strains. Different letters in (**B**,**D**) represent significant differences (*p* < 0.05) based on one-way ANOVA followed by post-hoc Tukey test.

**Figure 7 jof-10-00478-f007:**
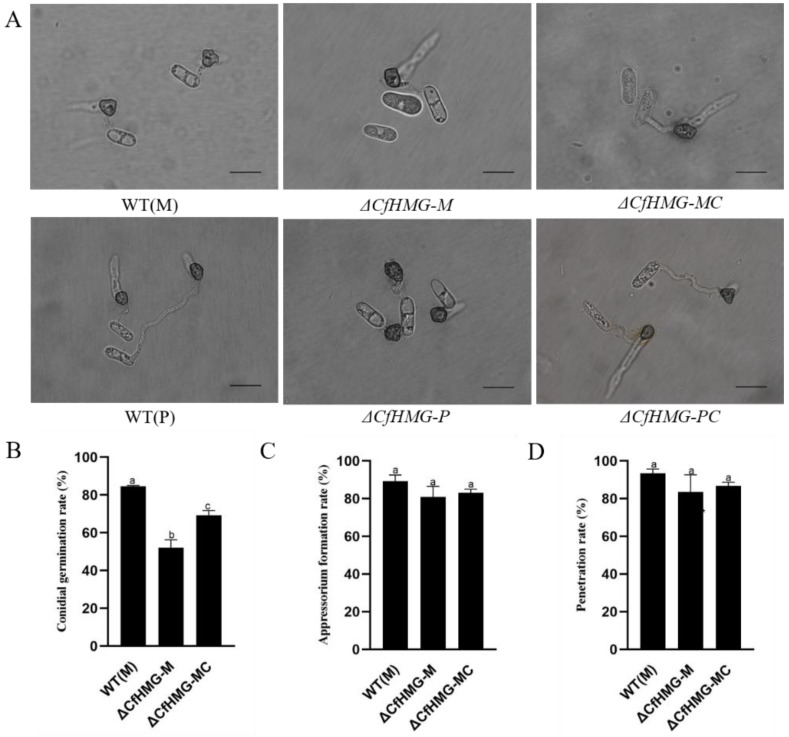
Defects of *CfHMG* mutants in conidial germination and formation of appressorium and penetration hypha. (**A**) Conidial germination and formation of appressoria and penetration hypha of indicated strains on cellophane at 12 hpi. Bar =20 μm. (**B**) The rate of conidial germination. (**C**) The rate of appressoria formation. (**D**) The rate of penetration hypha formation. Different letters in (**B**–**D**) represent significant differences (*p* < 0.05) based on one-way ANOVA followed by post-hoc Tukey test.

## Data Availability

The data presented in this study are openly available in FigShare at https://doi.org/10.1094/MPMI-11-19-0316-A.

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
