# Peer review of "CfHMG Differentially Regulates the Sexual Development and Pathogenicity of Colletotrichum fructicola Plus and Minus Strains"

_jof, 2024, doi:10.3390/jof10070478_

Round 1

Reviewer 1 Report

The authors demonstrated that MATA_HMG protein CfHMG in plus and minus Mating type strains of Colletotrichum fruticola is involved in sexual development during genetic crosses and influences virulence. The experimental work, analysis- and presentation of the results and interpretation of the results are suitable. However, the language proficiency in the text is inconsistent, with certain portions containing grammatical errors and imprecise expressions.

Furthermore, the authors should check the correct variety of the tested apple. Gala apple fruit is known by having a very distinctive yellow-orange skin with red striping. The apple shown on Figure 6 seems more of a Golden Delicious than a Gala (line 217 and Figure 6).

Here I commented only some of the sentences with wrong grammars or typos. The authors should ask the help of a native speaker to correct the whole text thoroughly.

Between lines 83 and 173 and

line 10: fruticola – fruticola (in italic)

line 10: different – difference

lines 12-14, 18-19, 19-21 and 21-23: sentences should be corrected grammatically

line 75: verb is missing from the sentence

The upstream and downstream flanking sequences of CfHMG with primers CfHMG-LFup/ CfHMG-LRup and CfHMG-RFDown/ 76 CfHMG-RRDown from gDNA.

lines 77-78: was is missing between” cassette constructed”

The gene replacement cassette constructed by connecting upstream and downstream flanking sequences with hygromycin resistance gene.

lines 80-81: The final PCR products were purified and transformed to  the wild type strains by PEG-mediated.

by PEG mediated method.

line 82: detect – detection

line 83: Two pair primers - Two pairs of primers

lines 174-175: “the wild type strain WT(P) produced aggregated perithecia since the 7 dpi.”

I do not understand “since the 7 dpi”

lines 197-198: wrong grammar

line 204: least - less

lines 205-206:

“In addition, in the cross of both plus and minus knock-out mutants, the size of perithecia was smaller.”

Smaller than what?

Author Response

Dear reviewer:

We are very grateful for your constructive comments and suggestions for our manuscript entitled "CfHMG differentially regulates sexual development and pathogenicity of Colletotrichum fructicola plus and minus strains" (Manuscript ID: jof-3046320). We have tried our best to improve and made some changes in the manuscript and the detailed corrections a listed below. In the revised manuscript, the revised parts based on reviewers’ suggestion or comments are highlighted in yellow, and those are added based on the newly supplied data illustration and revised by ourselves are highlighted in green.

Comments 1: The authors should check the correct variety of the tested apple. Gala apple fruit is known by having a very distinctive yellow-orange skin with red striping. The apple shown on Figure 6 seems more of a Golden Delicious than a Gala (line 217 and Figure 6).

Response 1: To see and show clearly the lesions on fruit infected, we used the bagged mature apple fruit, so it is in yellow. We did some revsion in Materials and Methods, as ”To see clearly the lesions on fruit infected, the bagged mature apple fruit in yellow were used. (p 3 line 22-23)”

Comments 2: However, the language proficiency in the text is inconsistent, with certain portions containing grammatical errors and imprecise expressions. Here I commented only some of the sentences with wrong grammars or typos. The authors should ask the help of a native speaker to correct the whole text thoroughly.

Response 2: Our co-author, the native English speaker has revised the whole text thoroughly.

line 11: fruticola – fruticola (in italic); line 11: different – difference

Response: Done.

lines 12-14, 18-19, 19-21 and 21-23: sentences should be corrected grammatically.

Response: Done.

line 75: verb is missing from the sentence.

Response: Done.

lines 77-78: was is missing between” cassette constructed”

Response: Revised.

lines 80-81: The final PCR products were purified and transformed to  the wild type strains by PEG-mediated.

Response: Revised as „The final PCR products were purified and transformed to the wild type strains by PEG-mediated protoplast transformation. (p 2 line 47)”

line 82: detect – detection

Response: Revised. (p 2 line 48).

line 83: Two pair primers

Response: We have rewritten the sentence in the revised manuscript.

lines 174-175: “the wild type strain WT(P) produced aggregated perithecia since the 7 dpi.”

Response: We have rewritten the sentence.

line 204: least – less

Response: Revised.

lines 205-206: In addition, in the cross of both plus and minus knock-out mutants, the size of perithecia was smaller.

Response: Revised. (p 7 line 16).

Dr. Guangyu Sun

Northwest A&F University
Yangling, China

Reviewer 2 Report

Overall, the work is well-structured, and this contribution should be considered for publication after addressing the following comments.

1. Re-write and organize the abstract, The abstract should be clear and concise, providing a brief overview of the study's purpose, methods, and findings.What methods were used to confirm the deletion of CfHMG in the mutants, and how was off-target activity ruled out? Discussing the broader implications of these findings, such as potential targets for controlling C. fructicola infections, would be beneficial

2. In section 2  conidia were filtrated with sterile MiraCloth 65 (EMD Millipore Corporation, USA), write in detail.

3. How might environmental factors, such as nutrient availability or temperature, influence the role of CfHMG in sexual development?

4. In Figure 3 Error bars came from three technical repetitions. Why the error bar is high for WT(m) similarly in figure 4 for (B,C, D)

5. The conclusion  part is missing,

Overall, the work is well-structured, and this contribution should be considered for publication after addressing the following comments.

1. Re-write and organize the abstract, The abstract should be clear and concise, providing a brief overview of the study's purpose, methods, and findings.What methods were used to confirm the deletion of CfHMG in the mutants, and how was off-target activity ruled out? Discussing the broader implications of these findings, such as potential targets for controlling C. fructicola infections, would be beneficial

2. In section 2  conidia were filtrated with sterile MiraCloth 65 (EMD Millipore Corporation, USA), write in detail.

3. How might environmental factors, such as nutrient availability or temperature, influence the role of CfHMG in sexual development?

4. In Figure 3 Error bars came from three technical repetitions. Why the error bar is high for WT(m) similarly in figure 4 for (B,C, D)

5. The conclusion  part is missing,

Author Response

Dear editors and reviewers,

We are very grateful for your constructive comments and suggestions for our manuscript entitled " CfHMG differentially regulates sexual development and pathogenicity of Colletotrichum fructicola plus and minus strains" (Manuscript ID: jof-3046320). Your comments are very valuable and helpful for improving our manuscript. In the revised manuscript, the revised parts based on reviewers’ suggestion or comments are highlighted in Yellow, and those are added based on the newly supplied data illustration and revised by ourselves are highlighted in green. In the following, the responses to all the comments are provided one by one.

  1. Re-write and organize the abstract, the abstract should be clear and concise, providing a brief overview of the study's purpose, methods, and findings. What methods were used to confirm the deletion of CfHMG in the mutants, and how was off-target activity ruled out? Discussing the broader implications of these findings, such as potential targets for controlling C. fructicola infections, would be beneficial

Response 1: We have reorganized the abstract.

  1. In section 2 conidia were filtrated with sterile MiraCloth 65 (EMD Millipore Corporation, USA), write in detail.

Response 2: Revised as ” To obtain conidia, three 6-mm-diameter mycelial plugs were placed in 60 ml of potato dextrose broth (PDB) to produce conidia with shaking at 180 rpm at 25°C for 4 d, after which culture liquids were filtered by three layers of sterile MiraCloth (EMD Millipore Corporation, USA). Centrifuge the filtrate for 3 min at 10,000 rpm and 4°C, and then discard the supernatant. ”p2 line 28-33.

  1. How might environmental factors, such as nutrient availability or temperature, influence the role of CfHMG in sexual development?

Response 3: We did the culturing of CfHMG mutants on PDA to observe the sexual development, however, there seems no difference between on PDA and OA. Thus in this manuscript, we did not show the related results on PDA.

  1. In Figure 3 Error bars came from three technical repetitions. Why the error bar is high for WT(m) similarly in figure 4 for (B, C, D)

Response 4: We recheck all data, and revised the related figures.

  1. The conclusion part is missing,

Response 5: Added.

Sincerely,

Dr. Guangyu Sun

Northwest A&F University
Yangling, China
